# Thrombotic Long-Term Consequences of SARS-CoV-2 Infection in Patients with Compensated Cirrhosis: A Propensity Score-Matched Analysis of a U.S. Database

**DOI:** 10.3390/diseases12070161

**Published:** 2024-07-17

**Authors:** Mark Ayoub, Carol Faris, Tajana Juranovic, Rafi Aibani, Morgan Koontz, Harleen Chela, Nadeem Anwar, Ebubekir Daglilar

**Affiliations:** 1Department of Internal Medicine, Charleston Area Medical Center, West Virginia University, Charleston, WV 25304, USA; tajana.juranovic@vandaliahealth.org (T.J.); rafi.aibani@vandaliahealth.org (R.A.); 2Department of Internal Medicine, Bayonne Medical Center, Bayonne, NJ 07002, USA; 3Health Services & Outcomes Research, CAMC-WVU Academic Medical Center, Charleston, WV 25304, USA; morgan.koontz@vandaliahealth.org; 4Division of Gastroenterology and Hepatology, Charleston Area Medical Center, West Virginia University, Charleston, WV 25304, USA; harleen.chela@camc.org (H.C.); nadeem.anwar@camc.org (N.A.)

**Keywords:** cirrhosis, SARS-CoV-2, COVID, PVT, DVT, PE, thrombosis, coagulation

## Abstract

Background: Cirrhosis causes an imbalance in the coagulation pathway and leads to a tendency for both bleeding and clotting. SARS-CoV-2 has been reported to be associated with a hypercoagulable state. This study examines SARS-CoV-2’s impact on hemostasis in compensated patients with cirrhosis. Methods: We analyzed the US Collaborative Network, which comprises 63 HCOs in the U.S.A. Compensated cirrhosis patients were split into two groups: SARS-CoV-2-positive and -negative. Patients’ baseline characteristics were used in a 1:1 propensity score-matched module to create comparable cohorts. We compared the risk of portal vein thrombosis (PVT), deep venous thrombosis (DVT), and pulmonary embolism (PE) at 6 months, and 1 and 3 years. Results: Of 330,521 patients, 27% tested positive and 73% remained negative. After PSM, both cohorts included 74,738 patients. Patients with SARS-CoV-2 had a higher rate of PVT compared to those without at 6 months (0.63% vs 0.5%, *p* < 0.05), 1 year (0.8% vs 0.6%, *p* < 0.05), and 3 years (1% vs. 0.7%, *p* < 0.05), a higher rate of DVT at 6 months (0.8% vs. 0.4%, *p* < 0.05), 1 year (1% vs. 0.5%, *p* < 0.05), and 3 years (1.4% vs. 0.8%, *p* < 0.05), and a higher rate of PE at 6 months (0.6% vs. 0.3%, *p* < 0.05), 1 year (0.7% vs. 0.4%, *p* < 0.05), and 3 years (1% vs. 0.6%, *p* < 0.05). Conclusions: The presence of SARS-CoV-2 infection in patients with compensated cirrhosis was associated with a higher rate of PVT, DVT, and PE at 6 months, and 1 and 3 years.

## 1. Introduction

As of February 2023, the global tally for confirmed SARS-CoV-2 cases has surpassed 754 million [1]. Among the myriad complications associated with SARS-CoV-2, venous thromboembolism (VTE) has emerged as a significant concern [2]. Previous research has predominantly honed in on VTE occurrences in hospitalized or severely ill SARS-CoV-2 patients, revealing a marked increase in risk compared to non-SARS-CoV-2 hospitalized individuals [3]. Recommendations have since leaned towards the administration of therapeutic doses of anticoagulation for certain hospitalized SARS-CoV-2 patients, with prophylactic doses advised for the remainder [4,5]. This guidance, however, does not address the majority who experience milder SARS-CoV-2 symptoms and manage their illness outside hospital settings.

The risk of VTE in non-hospitalized, or outpatient, adults with SARS-CoV-2 remains underexplored and somewhat contentious. Early studies present conflicting results; some suggest that VTE rates in SARS-CoV-2 outpatients are as low as 1.8 per 1000, aligning with general population rates, while others have reported rates as high as 18% among emergency department patients not requiring hospital admission [6,7,8]. The disparity in findings could be attributed to the studies’ reliance on small sample sizes or the use of administrative codes for VTE identification, which may not accurately reflect true VTE occurrences.

A notable clinical trial aimed at assessing the efficacy of prehospitalization prophylaxis for VTE in SARS-CoV-2 patients was prematurely concluded due to an event rate lower than anticipated [9]. This underscores the difficulties inherent in conducting prospective studies on VTE among outpatient SARS-CoV-2 cases. The challenge lies in accurately determining the actual risk of VTE, which is crucial for developing effective prevention and surveillance strategies for this population.

Cirrhosis significantly heightens the risk for thrombosis, such as portal vein thrombosis (PVT) and venous thromboembolism (VTE), and bleeding in individuals with chronic liver disease due to a dynamic imbalance between procoagulant and anticoagulant forces. This imbalance, fueled by the progression of cirrhosis, which diminishes the liver’s synthetic function and depletes natural anticoagulants, remains poorly understood. Contributing factors include reduced platelet production, enhanced platelet destruction due to hypersplenism, a decline in the synthesis of both Vitamin K-dependent and independent clotting and anticoagulant factors, and changes in purinergic signaling pathways [10]. This research is driven by the critical need to refine our understanding of SARS-CoV-2’s impact on cirrhosis-induced hemostatic alterations, with the ultimate goal of developing targeted prophylactic and therapeutic strategies that can mitigate the heightened risks of both clotting and bleeding in these patients.

## 2. Materials and Methods

Our study proposal was approved by the Institution Board Review Committee at the Charleston Area Medical Center (CAMC) with the IRB number 24-1076. We used the TriNetX database (located in Cambridge, MA, USA) for our analysis. TriNetX is a de-identified global and federal research network of real-time electronic medical records (EMR) data. Of the available healthcare networks available on TriNetX, we used the US Collaborative Network. The US Collaborative Network consists of 63 healthcare organizations (HCOs) located in the U.S.A. Due to the de-identified nature of the database, written informed consent from patients was waived. Adult patients aged ≥ 18 years with compensated cirrhosis were identified between March 2020 and March 2024. Patients with compensated cirrhosis were defined as patients who have documented diagnosis of cirrhosis using the corresponding International Classification of Diseases (ICD)-10 code. We included any patient with cirrhosis regardless of etiology. We excluded patients with any prior history of cirrhosis decompensation. Cirrhosis decompensation was defined as the presence of esophageal varices with or without bleeding, hepatic encephalopathy (HE), spontaneous bacterial peritonitis (SBP), jaundice, ascites, or hepatorenal syndrome (HRS). These conditions were also identified using the corresponding ICD-10 codes. A list of the ICD-10 codes used in the inclusion, exclusion, and propensity score matching (PSM) is provided in the Appendix A.

The included patients with compensated cirrhosis without any prior decompensation were divided into two groups: patients with compensated cirrhosis who tested positive for SARS-CoV-2 infection, and patients with compensated cirrhosis without any prior decompensation who tested negative for SARS-CoV-2 infection. Testing positive for SARS-CoV-2 infection was defined as the presence of IgG or IgM antibodies in the serum or the presence of SARS-CoV-2 RNA in the serum. Subsequently, we performed PSM between the two groups to ensure comparability and account for covariates. Following PSM, we compared different outcomes between the two groups. The outcomes that were studied were the risk of portal vein thrombosis (PVT), deep venous thrombosis (DVT), and pulmonary embolism (PE) at 6 months, 1 year, and 3 years. These outcomes were identified by their corresponding ICD-10 codes, as shown in the Appendix A. The difference between the two groups for all outcomes was calculated using Kaplan–Meier curves and log-rank tests. Additionally, the risk ratios (RRs), with 95% confidence intervals, were also calculated. Statistical significance was reported and defined with a *p*-value < 0.05. The previously mentioned methodology was performed using the built-in TriNetX platform, which is maintained by a multi-tenant Software as a Service (SaaS) system.

## 3. Results

### 3.1. Baseline Characteristics

A total of 330,521 patients with compensated cirrhosis from 2020 to 2024 met our inclusion criteria. Of those, 27% (*n* = 89,227) of patients with compensated cirrhosis tested positive for SARS-CoV-2 infection, and 73% (*n* = 241,294) of patients with compensated cirrhosis did not test positive for SARS-CoV-2 infection. Two propensity-matched cohorts were made, each comprising 74,738 patients. Propensity score matching included patients’ demographics, comorbidities, and medications.

The analysis of both cohorts’ baseline characteristics after propensity score matching did not show any significant difference between the two cohorts. Patients with compensated cirrhosis who tested positive for SARS-CoV-2 infection had a mean age of 58.8, with a standard deviation of 13.8. More than half (67%) of the cohort was white. Additionally, 45.8% of the cohort comprised females. In terms of comorbidities, 52.5% of patients with compensated cirrhosis who tested positive for SARS-CoV-2 infection were found to have hypertension. In addition, 12.8% had coronary artery disease (CAD). Chronic obstructive lung disease (COPD) was found in 10.9% of these patients, while chronic kidney disease (CKD) was found in 11.4%. Lastly, 3.3% of the patients had a history of stroke. In terms of patients’ medications, 26.7% of the patients with compensated cirrhosis that tested positive for SARS-CoV-2 infection were receiving anti-platelets, which included any of the following: ticlopidine, ticagrelor, aspirin, vorapaxar, cangrelor, clopidogrel, dipyridamole, or prasugrel. Additionally, 32.4% of the cohort were receiving an anticoagulant, which included any of the following: dabigatran, rivaroxaban, warfarin, desirudin, defibrotide, apixaban, argatroban, edoxaban, heparin peirudin, fondaparinux, bivalirudin, enoxaparin, dalteparin, tirofiban, or eptifibatide. More specifically, 1.8% of the cohort were on warfarin.

A full comparison of the cohorts’ baseline demographics and comorbidities before and after PSM is highlighted in Table 1.

### 3.2. Outcomes

After performing PSM between the two groups, we compared different outcomes. The rate of PVT was significantly higher in patients with compensated cirrhosis who tested positive for SARS-CoV-2 infection when compared to patients who did not test positive at 6 months (0.63% vs. 0.5%, *p* < 0.05). They also had a higher rate at 1 year (0.8% vs. 0.6%, *p* < 0.05), and 3 years (1% vs. 0.7%, *p* < 0.05). The rate of PVT over time is shown in Figure 1.

Similarly, the rate of DVT was significantly higher at 6 months in patients with compensated cirrhosis who tested positive for SARS-CoV-2 infection compared to patients who did not (0.8% vs. 0.4%, *p* < 0.05). Additionally, they had a higher rate of DVT at 1 year (1% vs. 0.5%, *p* < 0.05), and 3 years (1.4% vs. 0.8%, *p* < 0.05). The rate of DVT over time is shown in Figure 2.

Lastly, the patients with compensated cirrhosis who tested positive for SARS-CoV-2 infection, when compared to those who did not test positive, had a higher rate of PE at 6 months (0.6% vs. 0.3%, *p* < 0.05), 1 year (0.7% vs. 0.4%, *p* < 0.05), and 3 years (1% vs. 0.6%, *p* < 0.05). The rate of PE over time is shown in Figure 3.

A summary of all outcomes studied over time between the two groups is shown in Table 2.

## 4. Discussion

### 4.1. SARS-CoV-2 and Hypercoagulability

VTE and PE are known complications of different bacterial and viral infections, not only in patients hospitalized due to the presence of Wirchov’s triad but also in community settings in patients without additional risk factors, which suggests that infection can play a role as an independent risk factor [11]. A retrospective study performed over 13 years among residents in Minnesota with objectively diagnosed DVT and PE (cases, *n* = 1303) showed that 39.4% of patients with VTE had experienced an infection in the previous 92 days [12]. Additional analysis showed that the infections with the highest risk were pneumonia, symptomatic UTI, and oral, intrabdominal, and bloodstream infections [12]. The insults caused by SARS-CoV-2 have been related to the hyperinflammatory state caused by the virus [13]. The main role in pathophysiology is endothelial injury and cytokine storm causing transient hypercoagulability and a state called thromboinflammation. Elevated fibrinogen levels and increased thrombin formation with endothelial injury cause decreased levels of circulating anticoagulants, including protein C, S, and antithrombin III. As the process progresses, coagulation factors and platelets are consumed, leading to a potentially fatal disseminated intravascular coagulation (DIC) [14].

SARS-CoV-2 is a contagious respiratory virus whose clinical spectrum can range from no symptoms to mild pneumonia to adult respiratory distress syndrome (ARDS) secondary to a cytokine storm. Since the beginning of the SARS-CoV-2 era, multiple authors have reported an association between SARS-COV-2 and PE without other risk factors for VTE and severe forms of the disease, suggesting that SARS-CoV-2 represents an independent risk factor [15,16,17]. The pathophysiology of coagulopathy in SARS-CoV-2 is similar to the coagulopathy associated with other infections, but the concept of thrombosis is different. After entering the nasopharynx and lungs, SARS-CoV-2 binds to the angiotensin-converting enzyme 2 receptors on the vascular endothelium, activating the local and systemic inflammatory response. This has the potential to cause lung injury and ARDS with characteristic microvascular thrombosis found postmortem rather than thrombosis caused by dislodged thrombus from deep venous thrombosis. SARS-CoV-2 will characteristically cause this microthrombosis without systemic endothelial injury and with only localized inflammation and endothelial apoptosis [18]. Studies on SARS-CoV-2 patients have demonstrated PE in central pulmonary artery locations (main and lobar pulmonary arteries) as well as in segmental and subsegmental pulmonary arteries [19]. Available data on PE in SARS-CoV-2 show PE in 20–30% of those infected with SARS-CoV-2 [20,21]. A study published in February 2023 on 193 patients showed an incidence of 21.8%, with no associations between PE and inflammatory markers (IL-6, CRP, troponin, LDH, and ferritin) or lymphopenia. Only elevated D-dimer showed a statistically significant association with PE, which is expected, given the pathophysiology of thromboinflammation [20]. Traditional risk factors, such as age, diabetes, hypertension, prior VTE, and smoking, are not associated with a higher incidence of PE in SARS-CoV-2 [9]. A meta-analysis of 21 studies published in 2021 showed similar results. There was no association between traditional risk factors in PE in SARS-CoV-2, but only moderate certainty evidence between D-dimer and PE [22]. Recent data show that the risk of PE and DVT remains elevated up to 70 days after SARS-CoV-2 infection for DVT and up to 110 days for PE. Furthermore, the highest event ratio was recorded among the critically ill during the first wave of the SARS-CoV-2 pandemic, compared with the second and third waves [23]. Despite all this information, there is not enough evidence to recommend thromboembolic prophylaxis for nonhospitalized patients, only for hospitalized patients with SARS-CoV-2 [23].

#### 4.1.1. DVT and PE

SARS-CoV-2 has been linked to increased blood clotting tendencies. Infection-induced endothelial cell dysfunction leads to higher levels of thrombin production and a shutdown in fibrinolysis. Some studies have noted significant hematological lab abnormalities in patients with both arterial and venous thrombosis [24]. Severe SARS-CoV-2 cases with hypoxia can further promote thrombus formation via a hypoxia-inducible transcription factors (HIFs) signaling pathway, whose target genes include factors that promote thrombus formation [25]. Additionally, a signaling pathway that is dependent on hypoxia-inducible transcription factors contributes to heightened blood coagulation [26].

Procoagulant clotting abnormalities and thromboembolism are increasingly being recognized as common complications among critically ill SARS-CoV-2 patients. These complications might play a role in exacerbating both morbidity and mortality rates [27]. In a retrospective analysis conducted at two French intensive care units (ICUs) involving 26 SARS-CoV-2 patients, a total of 18 cases (69%) of peripheral venous thromboembolism (VTE) were documented, with 6 patients (23%) diagnosed with pulmonary embolism. Notably, some patients who received therapeutic anticoagulation upon admission still experienced VTE, underscoring the prothrombotic nature of SARS-CoV-2 [28].

An analysis of the National Inpatient Sample (NIS) compared in-hospital outcomes for patients with deep vein thrombosis (DVT) and/or pulmonary embolism (PE) along with concurrent SARS-CoV-2 infection to those with concurrent influenza infection, across 62,895 hospitalizations. The findings showed increased rates of cardiac arrest and inpatient mortality among SARS-CoV-2 patients. This discovery indirectly suggests a potential link between DVT, PE, and the severity of SARS-CoV-2 infection [29] A study by Panigada et al. indicated a higher likelihood of venous thrombosis in patients with SARS-CoV-2 [26]. According to a meta-analysis of 28 trials, there is no association between prophylactic and therapeutic anticoagulation in SARS-CoV-2 infection and VTE occurrences. One explanation could be that inflammation and hypercoagulability are probably linked to the pathophysiology of VTE and additional anti-inflammatory therapy may prove superior to the AC alone [29]. This further highlights the hypercoagulability associated with SARS-CoV-2.

#### 4.1.2. PVT

Taquet et al. conducted their study primarily in the United States, employing a retrospective cohort analysis using electronic health records. They investigated the absolute risks of cerebral venous thrombosis (CVT) and portal venous thrombosis (PVT) in 537,913 confirmed SARS-CoV-2 cases during the two weeks following infection. This was compared with other cohorts, including influenza patients and individuals who had received an mRNA vaccination for SARS-CoV-2. The study found that the incidence of CVT and PVT after SARS-CoV-2 was significantly higher than in matched control cohorts. Specifically, in the two weeks following SARS-CoV-2 infection, the risk of PVT was significantly higher when compared to a matched cohort who received an mRNA vaccination with a relative risk of 4.5 [30].

Hassnine et al. conducted a cross-sectional, observational controlled study involving 70 individuals with liver cirrhosis, who were divided into two groups matched for age and sex. Group A comprised 28 individuals with liver cirrhosis and SARS-CoV-2, while Group B consisted of 42 individuals with hepatic cirrhosis alone as controls. In Group A, PVT was detected in three cases (10.7%), all of which were previously undiagnosed. In contrast, Group B (liver cirrhosis only) had one patient (2.3%) diagnosed with PVT. This difference was statistically significant with a *p*-value of less than 0.05, indicating a higher prevalence of PVT in the liver cirrhosis and SARS-CoV-2 group compared to the liver cirrhosis-only group [31].

Muñoz et al. conducted a retrospective study on 1127 patients admitted to the Infanta Leonor University Hospital [32]. They found that 6.1% of patients experienced thrombotic events, which is equivalent to 80 events, occurring in 69 individuals, all of whom were diagnosed with SARS-CoV-2 infection [32]. Venous thromboembolism (VTE) was the most common type of thrombosis, affecting 71% of patients (49 out of 69) and constituting 65% of the events (52 out of 80). These events included 44 cases of pulmonary embolism (PE), 6 cases of deep vein thrombosis (DVT), and 2 cases of portal vein thrombosis (PVT). Two patients, a 27-year-old male and a 67-year-old female, were diagnosed with PVT and treated with enoxaparin before being discharged. Notably, despite receiving prophylactic treatment, 90% of the patients in this study experienced a thrombotic episode [32,33]. In the PROTHROMCOVID randomized controlled trial, thrombotic events still occurred in patients receiving both prophylactic and therapeutic doses of anticoagulation, shedding more light on the thrombotic properties of the SARS-CoV-2 virus [33].

### 4.2. SARS-CoV-2 and Bleeding

A multicenter retrospective study in 400 admitted patients with SARS-CoV-2 infection in 2020 highlighted an overall bleeding rate of 4.8% and major bleeding rates of 2.3% [34]. The bleeding events were mostly gastrointestinal. Another study reported bleeding to be the cause of death in 6% of patients with SARS-CoV-2 infection [35]. Furthermore, these findings were supported in a meta-analysis by Jiménez et al., who found a 7.8% pooled incidence of bleeding and a 3.9% pooled incidence of major bleeding events [36]. Another study found that 40% of patients with SARS-CoV-2 pneumonia that had GI bleed had stool that was PCR positive for SARS-CoV-2 [37]. Of interest, in patients with compensated cirrhosis who tested positive for SARS-CoV-2, bleeding was reported in 1.3% of the patients at 1 month from testing positive [11]. Although bleeding is relatively more uncommon than thrombosis in patients with SARS-CoV-2 infection, it is still reported and warrants evaluation to better understand the long-term effects [38].

### 4.3. Cirrhosis and Coagulation

Patients with end-stage liver disease (ESLD) are not only known to be at risk for bleeding but also risk of thrombosis. The cause of hypercoagulability is multifactorial, but endothelial dysfunction (ED) induced by inflammation and oxidative stress plays a main role. Patients with liver cirrhosis have persistent inflammation and low-level endotoxemia due to decreased portal clearance of gut bacteria. Persistent endotoxemia stimulates TNF-alfa production, which leads to ED, increased levels of vWF, FVIII, and PAI synthesized in the endothelium, and platelet hyperactivity, promoting thrombosis. Additional factors for thrombosis are decreased levels of protein C, S, and ATIII in liver disease. Some populations with ESLD, like those with NASH, autoimmune hepatitis, and chronic hepatitis C, and the African American race, are more susceptible to thrombosis [39].

In patients with cirrhosis, especially those with hepatocellular carcinoma (HCC), there may be a link between increased blood clotting tendencies and a perceived higher risk of developing portal vein thrombosis. Portal vein thrombosis (PVT) is not commonly found in the general population. According to a study conducted during autopsies, which covered 84% of all deaths in hospitals in Malmo, the overall lifetime occurrence of PVT is 1.0%. However, 40% of individuals in the study also had cirrhosis, and those with both liver disease and hepatocellular carcinoma (HCC) were at the greatest risk of developing PVT [40].

The coagulation balance pathways in patients with cirrhosis are unpredictable. Natural disease progression affects both anti- and procoagulant pathways, which results in an imbalanced coagulation system that may favor thrombosis [41]. Advancing liver disease includes decreased protein C and antithrombin and increased endothelial-derived von Willebrand factor and factor VII [42].

### 4.4. SARS-CoV-2 in Cirrhosis

SARS-CoV-2 causes vascular endothelial destruction, which causes vasoconstriction and subsequent procoagulant state [43,44]. Spiezia reported elevated fibrinogen and D-dimer levels, indicating a procoagulant state [45]. This was further confirmed by a systematic review that found a higher association rate of PVT in hospitalized patients with SARS-CoV-2 infection [46]. PVT is a well-known cause of hepatic decompensation, which in turn plays a role in the coagulation imbalance [41,47,48]. Additionally, the presence of SARS-CoV-2 infection in patients with previously compensated cirrhosis who did not have any prior decompensation was found to be an independent risk factor for hepatic decompensation in those patients [11].

### 4.5. Summarization

The previously mentioned studies show a controversial role of SARS-CoV-2 infection on the hemostatic pathways. Many studies show an inclination for thrombosis in patients with SARS-CoV-2 infection, while other studies show an inclination for bleeding tendency [49]. Patients with cirrhosis have an imbalanced coagulation pathway that hinders the use of regular lab values as a reliable marker for coagulability [41]. The presence of cirrhosis causes a dynamic disequilibrium between procoagulant and anticoagulant states in those patients which makes their hemostatic outcomes unpredictable [10]. This raises the importance of assessing outcomes in patients with cirrhosis who have SARS-CoV-2 to better understand the effect of SARS-CoV-2 on their hemostatic pathways. This highlights the importance of our study, which shows a tendency for thrombosis in patients with cirrhosis who have SARS-CoV-2 infection. This sheds light on their response to the infection and its outcomes.

### 4.6. Strengths and Limitations

Our study outcomes align with other studies that show a higher risk of thrombosis in patients with SARS-CoV-2, specifically, PVT, DVT, and PE. One of our major study strengths is the large sample of patients included through a national database, which allows national generalizability within the USA, in addition to the use of PSM to account for variables such as the use of anti-platelets or anticoagulants to ensure comparable cohorts.

Our study does not come without limitations. First, our database mainly comprised patients treated within the American healthcare system, thereby limiting the generalizability of our findings outside of the U.S.A. Second, despite the use of PSM to account for relevant variables, the de-identified nature of our database and the retrospective nature of our study still leave room for possible misclassification bias and residual confounding that is unaccounted for. Third, our study did not account for SARS-CoV-2 treatments, such as corticosteroid exposure, which have high prothrombotic potential as well as a higher risk for sepsis which increases the risk of thrombosis [50]. Lastly, the nature of our de-identified retrospective database did not allow us to characterize the severity of the patients’ liver disease using calculators such as the Model for End-Stage Liver Disease (MELD) score or the Child–Pugh classification; however, we attempted to mitigate this limitation by excluding any patient with decompensated cirrhosis and by using PSM.

## 5. Conclusions

Patients with cirrhosis have a dynamic disequilibrium between procoagulant and anticoagulant states, which increases their tendency for thrombosis and bleeding, respectively. Current studies on the effect of SARS-CoV-2 infection show a higher tendency for thrombosis but also report a bleeding tendency as well. This underscores the importance of understanding the long-term effects of SARS-CoV-2 infection in patients with cirrhosis. Our study shows that the presence of SARS-CoV-2 infection in patients with compensated liver cirrhosis was associated with a higher rate of PVT, DVT, and PE at 6 months, and 1 and 3 years. This sheds light on the long-term hematological and thrombotic effect of SARS-CoV-2 in this population.

## Figures and Tables

**Figure 1 diseases-12-00161-f001:**
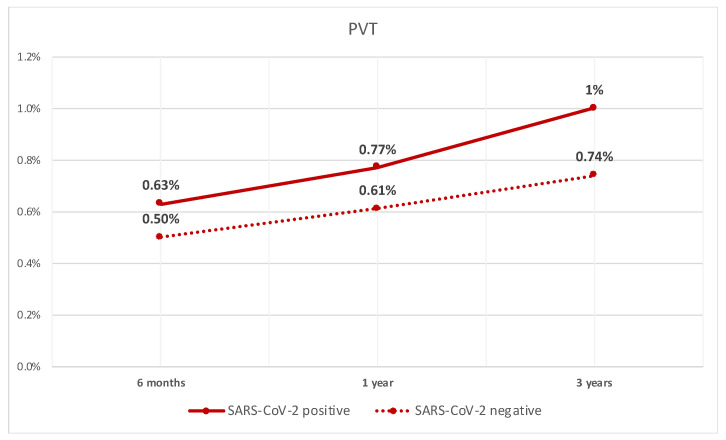
Line graph of rate of PVT in patients with compensated cirrhosis with SARS-CoV-2 infection and those without.

**Figure 2 diseases-12-00161-f002:**
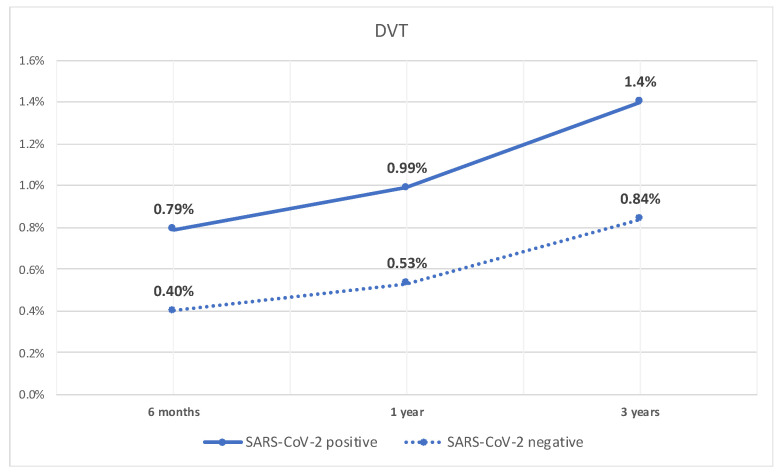
Line graph of rate of DVT in patients with compensated cirrhosis with SARS-CoV-2 infection and those without.

**Figure 3 diseases-12-00161-f003:**
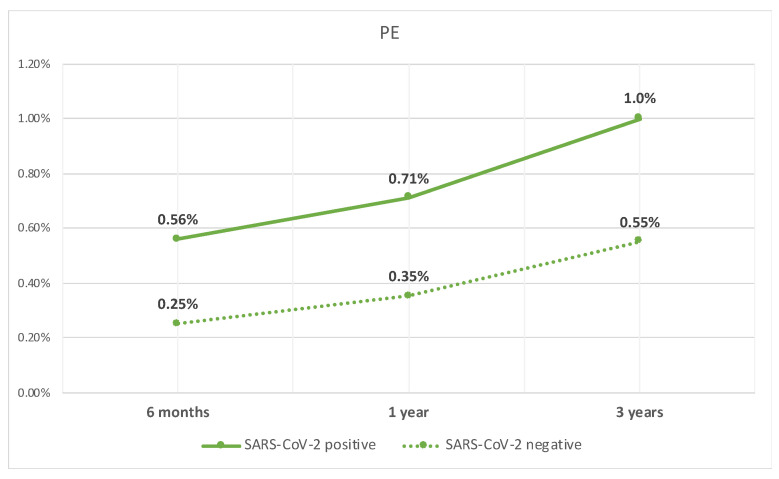
Line graph of rate of PE in patients with compensated cirrhosis with SARS-CoV-2 infection and those without.

**Table 1 diseases-12-00161-t001:** Patients’ baseline characteristics before and after PSM.

Characteristic	SARS-CoV-2 Positive *n* = 89,227	SARS-CoV-2 Negative *n* = 241,294	*p*-Value	SARS-CoV-2 Positive *n* = 74,738	SARS-CoV-2 Negative *n* = 74,738	*p*-Value
Demographics						
Age at Index (Mean ± SD)	59.5 ± 13.7	57.5 ± 13.3	<0.001	58.8 ± 13.8	58.6 ± 13.6	0.192
Female	46.3%	43.0%	<0.001	45.8%	45.5%	0.287
White	67.1%	62.6%	<0.001	67.0%	67.2%	0.352
Black or African American	15.3%	11.3%	<0.001	14.5%	14.2%	0.229
Diagnosis						
CAD	17.0%	5.6%	<0.001	12.8%	12.7%	0.281
CKD	15.3%	5.1%	<0.001	11.4%	11.3%	0.396
COPD	14.3%	5.0%	<0.001	10.9%	10.8%	0.855
Hypertension	59.5%	26.9%	<0.001	52.5%	54.5%	0.092
Diabetes mellitus	35.5%	16.2%	<0.001	30.7%	32.5%	0.067
Medication						
Anti-platelets	33.8%	12.1%	<0.001	26.7%	27.0%	0.316
Anticoagulants	42.9%	12.2%	<0.001	32.4%	31.6%	0.073
Warfarin	2.1%	0.9%	<0.001	1.8%	1.7%	0.568

**Table 2 diseases-12-00161-t002:** Summary of outcomes at 6 months, 1 year, and 3 years.

	PVT			DVT			PE		
	SARS-CoV-2 *n* = 74,738	No SARS-CoV-2 *n* = 74,738	*p*-Value	SARS-CoV-2 *n* = 74,738	No SARS-CoV-2 *n* = 74,738	*p*-Value	SARS-CoV-2 *n* = 74,738	No SARS-CoV-2 *n* = 74,738	*p*-Value
6 months	470 (0.63%)	372 (0.50%)	0.001	592 (0.79%)	297 (0.40%)	<0.001	418 (0.56%)	187 (0.25%)	<0.001
1 year	577 (0.77%)	454 (0.61%)	<0.001	739 (0.99%)	397 (0.53%)	<0.001	533 (0.71%)	258 (0.35%)	<0.001
3 years	732 (1.0%)	553 (0.74%)	0.000	1027 (1.37%)	630 (0.84%)	0.000	749 (1.0%)	413 (0.55%)	0.000

## Data Availability

Available data are presented. Additional data are only available as permitted by a third party.

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
