# Peer review of "Thrombotic Long-Term Consequences of SARS-CoV-2 Infection in Patients with Compensated Cirrhosis: A Propensity Score-Matched Analysis of a U.S. Database"

_diseases, 2024, doi:10.3390/diseases12070161_

Round 1

Reviewer 1 Report

Comments and Suggestions for Authors

The subject of the article is an interesting one, approaching the long-term effects of Sars Cov2 infection in cirrhotic patients. The manuscript respects the structure of an original article.

The authors did not mention the limitations and the strengths of the study and they must be done. 

Author Response

--Thank you so much for your time, effort and valuable feedback!

The subject of the article is an interesting one, approaching the long-term effects of Sars Cov2 infection in cirrhotic patients. The manuscript respects the structure of an original article.

The authors did not mention the limitations and the strengths of the study and they must be done. 

--Thank you for bringing this to our attention. We added paragraph 4.6. at the end stating our strengths and limitations!

Reviewer 2 Report

Comments and Suggestions for Authors

I have truly enjoyed reading the paper investigating long-term thrombotic complications in patients with compensated liver cirrhosis in association with COVID-19. The paper is well written and it is bringing forward an important issue of long-term thrombotic consequences of systemic inflammatory disease such as COVID-19. Although it is hard to directly associated thrombosis that is more than 6 months or several years away from the infectious insult, it should be noted that probably patients more susceptible to COVID-19 are also having more prothrombotic liver disease. There are only some minor points that I would like to mention:

1) I suggest to change the title to state thrombotic instead of hematological

2) in abstract it all abbreviations should be explained, especially PVT.

3) are there data on corticosteroid exposure of included patients, since these drugs are typically first line of severe COVID treatment, as well as have high prothrombotic potential. If not please state this as a limitation of the paper, also comment on possible septic complications that may be potentiated by steroid use in patients with cirrhosis and COVID-19 (doi:  10.3390/v16010086). 

Author Response

--Thank you so much for your time and feedback that allows us to enhance our manuscript! 

I have truly enjoyed reading the paper investigating long-term thrombotic complications in patients with compensated liver cirrhosis in association with COVID-19. The paper is well written and it is bringing forward an important issue of long-term thrombotic consequences of systemic inflammatory disease such as COVID-19. Although it is hard to directly associated thrombosis that is more than 6 months or several years away from the infectious insult, it should be noted that probably patients more susceptible to COVID-19 are also having more prothrombotic liver disease. There are only some minor points that I would like to mention:

1. I suggest to change the title to state thrombotic instead of hematological

--That is an excellent idea and suits the article better. We went ahead and changed the title to “Thrombotic Long-Term Consequences of SARS-CoV-2 Infection in Patients with Compensated Cirrhosis: A Propensity-Score Matched Analysis of a U.S. Database”

2. in abstract it all abbreviations should be explained, especially PVT.

--Thank you for highlighting that. We went ahead and wrote all the abbreviations in the abstract.

3. are there data on corticosteroid exposure of included patients, since these drugs are typically first line of severe COVID treatment, as well as have high prothrombotic potential. If not please state this as a limitation of the paper, also comment on possible septic complications that may be potentiated by steroid use in patients with cirrhosis and COVID-19 (doi:  10.3390/v16010086). 

--Excellent point! Since we were looking at outcomes up to 3 years, we unfortunately, did not account for SARS-CoV-2 treatment. We added this to our study limitations in paragraph 4.6. and referenced the mentioned study.

Reviewer 3 Report

Comments and Suggestions for Authors

Most importantly, based on the iThenticate report, it seems that the methods are identical to another study, however, this is not supported by any references. 

The discussion is disorganized, as it keeps going back and forth about how COVID-19 may cause thrombosis and how to prevent it, multiple times. 

Line 166 - need a reference for the study mentioned. 

Line 184 - not every Sars-CoV-2 case ends up in ARDS. 

Line 213 - "thickening the blood" - is not a scientific term. 

Line 241 - The number 59,7913 seems to be an error. 

The Discussion section is just a simple list of previous studies and their findings, but not a true discussion. This is without discussing how those findings are relevant to the current study and why? 

Line 263 - there is no such thing as "clinically diagnosed with SARS" - it has to be confirmed by at least one accepted laboratory method. 

The Conclusions section is insufficient. 

Author Response

--Thank you for the valuable feedback! We appreciate your time and effort you put in reviewing our manuscript which allows us to enhance and improve our article.

Most importantly, based on the iThenticate report, it seems that the methods are identical to another study, however, this is not supported by any references. 

--Thank you for highlighting this. Our study methodology is similar to another study we performed as it is using the same database. We added the reference to reflect that.

The discussion is disorganized, as it keeps going back and forth about how COVID-19 may cause thrombosis and how to prevent it, multiple times. 

--We wanted to highlight the controversial role of SARS-CoV-2 on the coagulation cascade and how some studies found a pro-thrombotic and how other studies found a bleeding risk with the infection. This is to shed light on such effect and highlight the importance of better understanding of the mechanism specially in patients with cirrhosis who have an unpredictable hemostatic pathway. This is also to further showcase our study findings that shows a pro-thrombotic effect. This is fully described in paragraphs 4.1. and 4.2. We also added a summarization paragraph 4.5. to reflect our thought process.

Line 166 - need a reference for the study mentioned. 

--Thank you for highlighting this. We added the study reference.

Line 184 - not every Sars-CoV-2 case ends up in ARDS. 

--Thank you for your comment. We rephrased the sentence showcasing the potential of ARDS with SARS-CoV-2 infection.

Line 213 - "thickening the blood" - is not a scientific term. 

--Thank you for highlighting this. We rephrased the sentence and added the study reference.

Line 241 - The number 59,7913 seems to be an error. 

--Thank you for highlighting the error. We fixed the number.

The Discussion section is just a simple list of previous studies and their findings, but not a true discussion. This is without discussing how those findings are relevant to the current study and why? 

--Thank you for bringing this to our attention. We added paragraph 4.5. to summarize the findings of those studies in relation to ours and why our study is relevant.

Line 263 - there is no such thing as "clinically diagnosed with SARS" - it has to be confirmed by at least one accepted laboratory method. 

--Thank you for highlighting this. You are correct, those patients were diagnosed by SARS-CoV-2 positive reverse transcriptase-polymerase chain reaction [RT-PCR] of nasopharyngeal swabs or isolation of SARS-CoV-2 in a clinical specimen. We changed the phrasing to match the way they reported their findings and added the reference.

The Conclusions section is insufficient. 

--Thank you for highlighting this. We expanded on our discussion section.

--Again, thank you for the valuable feedback!

Round 2

Reviewer 3 Report

Comments and Suggestions for Authors

The authors made significant improvements to the manuscript, with some parts completely re-written. 

Specific comments for the new version: 

Line 177: Sars-CoV-2 is not an illness. It is a virus. 

Line 361: What is meant by "Out study" ? 

Comments on the Quality of English Language

No major issues seen. 

Author Response

The authors made significant improvements to the manuscript, with some parts completely re-written. 

Specific comments for the new version: 

Line 177: Sars-CoV-2 is not an illness. It is a virus. 

Thank you for pointing that out. We rephrased the line.

Line 361: What is meant by "Out study" ? 

Thank you for pointing that out. This is a typo- "Our study". We corrected the spelling error.

Thank you again for your diligence and invaluable feedback!